# SARS-CoV-2 Vaccine Willingness among Pregnant and Breastfeeding Women during the First Pandemic Wave: A Cross-Sectional Study in Switzerland

**DOI:** 10.3390/v13071199

**Published:** 2021-06-22

**Authors:** Sarah Stuckelberger, Guillaume Favre, Michael Ceulemans, Hedvig Nordeng, Eva Gerbier, Valentine Lambelet, Milos Stojanov, Ursula Winterfeld, David Baud, Alice Panchaud, Léo Pomar

**Affiliations:** 1Department Woman-Mother-Child, Lausanne University Hospital and University of Lausanne, 1011 Lausanne, Switzerland; Sarah.Stuckelberger@chuv.ch (S.S.); guillaume.favre@chuv.ch (G.F.); Eva.Gerbier@chuv.ch (E.G.); valentine.lambelet@gmail.com (V.L.); Milos.Stojanov@chuv.ch (M.S.); David.Baud@chuv.ch (D.B.); 2Department of Pharmaceutical and Pharmacological Sciences, KU Leuven, 3000 Leuven, Belgium; michael.ceulemans@kuleuven.be; 3Teratology Information Service, Pharmacovigilance Centre Lareb, 5237 MH ‘s-Hertogenbosch, The Netherlands; 4Pharmacoepidemiology and Drug Safety Research Group, Department of Pharmacy, PharmaTox Strategic Initiative, Faculty of Mathematics and Natural Sciences, University of Oslo, 0315 Oslo, Norway; h.m.e.nordeng@farmasi.uio.no; 5Department of Child Health and Development, Norwegian Institute of Public Health, 0403 Oslo, Norway; 6Swiss Teratogen Information Service, Service de Pharmacologie Clinique, Lausanne University Hospital and University of Lausanne, 1011 Lausanne, Switzerland; Ursula.Winterfeld@chuv.ch; 7Institute of Primary Health Care (BIHAM), University of Bern, 3012 Bern, Switzerland; 8Service of Pharmacy, Lausanne University Hospital and University of Lausanne, 1011 Lausanne, Switzerland; 9School of Health Sciences (HESAV) Midwifery Department, University of Applied Sciences and Arts Western Switzerland, 1011 Lausanne, Switzerland

**Keywords:** SARS-CoV-2, coronavirus, COVID-19, pregnancy, breastfeeding, vaccine willingness

## Abstract

As pregnant women are at high risk of severe SARS-CoV-2 infection and COVID-19 vaccines are available in Switzerland, this study aimed to assess the willingness of Swiss pregnant and breastfeeding women to become vaccinated. Through a cross-sectional online study conducted after the first pandemic wave, vaccination practices and willingness to become vaccinated against SARS-CoV-2 if a vaccine was available were evaluated through binary, multi-choice, and open-ended questions. Factors associated with vaccine willingness were evaluated through univariable and multivariable analysis. A total of 1551 women responded to questions related to the primary outcome. Only 29.7% (153/515) of pregnant and 38.6% (400/1036) of breastfeeding women were willing to get vaccinated against SARS-CoV-2 if a vaccine had been available during the first wave. Positive predictors associated with SARS-CoV-2 vaccine acceptance were an age older than 40 years, a higher educational level, history of influenza vaccination within the previous year, having an obstetrician as the primary healthcare practitioner, and being in their third trimester of pregnancy. After the first pandemic wave, Switzerland had a low SARS-CoV-2 vaccination acceptance rate, emphasizing the need to identify and reduce barriers for immunization in pregnant and breastfeeding women, particularly among the youngest and those with a lower educational level.

## 1. Introduction

In 2020, the outbreak of a novel coronavirus, the severe acute respiratory syndrome coronavirus 2 (SARS-CoV-2), was declared a pandemic with more than 166 million confirmed cases worldwide. In Switzerland, more than 680,000 people tested positive with more than 10,000 deaths reported [1].

Pregnant women are considered a vulnerable population for SARS-CoV-2 infection. Current evidence suggests that they are up to 70% more susceptible to infection. If infected, they are also at greater risk of developing complications [2,3,4] such as admission to an intensive care unit, mechanical ventilation, and death [5,6]. Increased risk of caesarian section, iatrogenic prematurity, post-partum hemorrhage, preeclampsia, and miscarriage have also been reported [7,8,9,10,11].

Currently, two SARS-CoV-2 mRNA vaccines approved by Swissmedic (the Swiss authority for the utilization and surveillance of therapeutic products) are used in the vaccine campaign in Switzerland [12]. However, vaccines cannot curb epidemics without widespread acceptance. The World Health Organization (WHO) has listed vaccine hesitancy as one of the top ten threats to global health [13], especially for populations at risk. In Switzerland, as in many countries, vaccination programs have already been established to protect pregnant women and their infants from serious infections such as influenza and pertussis. Both influenza and pertussis vaccines have proven to be effective in protecting mothers and their newborns [14,15]. However, immunization rates for influenza and pertussis have been disappointingly low in Switzerland [16] mainly due to a lack of adequate promotion and compliance [17]. Low uptake of vaccination in pregnancy has been reported worldwide [18,19] with several studies identifying inadequate knowledge about the disease threat; doubts about vaccine safety, efficacy, and benefits; and the lack of recommendations from vaccine providers, as the main obstacles among pregnant women [20,21,22]. Maternal characteristics may also play a role. Unemployment, younger age (<25 years old), and high perceived stress have been associated with lower vaccination rates during pregnancy, whereas a history of depression increased the likelihood of being vaccinated [23].

SARS-CoV-2 vaccination has recently been recommended in Switzerland for pregnant women who have additional risk factors or are at high risk of exposure through their work. This vaccination strategy may represent a barrier to the successful vaccination of all members of this high-risk group, especially when compared to some countries where pregnant women are routinely vaccinated or considered a priority group. This is a glaring example of the need to better understand the many factors influencing the acceptance of and access to vaccination, especially among more vulnerable populations such as pregnant women to develop targeted information campaigns.

Thus, in a cross-sectional survey during the first wave of the pandemic, we investigated COVID-19 vaccine willingness among Swiss pregnant and breastfeeding women if a vaccine was available, as well as the factors contributing to their acceptance or hesitancy.

## 2. Materials and Methods

### 2.1. Study Population and Data Collection

This Swiss cross-sectional online study is part of a European multi-center study conducted in several countries (Belgium, Ireland, Norway, The Netherlands, United Kingdom) and approved by the Ethics Committee Research of UZ/KU Leuven (id: S63966). The questionnaire used in Switzerland was available in German, French, and Italian. The goal was to examine the overall impact of the SARS-CoV-2 pandemic on pregnant and breastfeeding women (i.e., pregnancy/breastfeeding experience, life and professional habits, mental health status, relationship with the healthcare system, medication use, and vaccine perceptions during pregnancy/breastfeeding) [24]. The COVID-19 vaccine willingness of pregnant and breastfeeding women included in the multi-center study has already been published [25], and the Swiss rate was among the lowest, hence the need to investigate the factors associated with vaccine acceptance in a Swiss-specific study.

In Switzerland, the online questionnaire was accessible from 18 June to 12 July 2020 through websites, forums, and social media (www.letsfamily.ch, www.swissmom.ch, www.medela.ch, www.chuv.ch). All data were collected and processed anonymously. All participants provided online informed consent prior to survey initiation.

### 2.2. Study Population

To be eligible, Swiss women needed to be at least 18 years old and be pregnant at the time of the survey or have breastfed within the past three months.

### 2.3. Variables

We collected information on sociodemographic characteristics (i.e., age, primary language, marital status, working status, education level), medical history (i.e., gravidity, parity, co-morbidities, smoking during pregnancy, main practitioner for the pregnancy follow-up, clinical course of the neonate for breastfeeding mothers), exposure to SARS-CoV-2 or presence in an at-risk setting (i.e., symptoms potentially related to COVID-19, hospitalization related to COVID-19, testing by RT-PCR, serology or computed tomography, living with someone who tested positive, co-habiting with an elderly person (>65 years old)). The negative impact of the SARS-CoV-2 pandemic on the pregnancy/breastfeeding experience, life habits, and work was assessed through participants graded answers: “yes” or “rather yes”, grouped as “negative impact of the SARS-CoV-2 pandemic”; and “rather no” or “no”, grouped as “no negative impact of the SARS-CoV-2 pandemic”. Mental health status was assessed using validated screening tests including the Edinburgh Postnatal Depression Scale for depression [26,27], the Generalized Anxiety Disorder 7-item Scale for anxiety [28], and the Perceived Stress Scale for stress [29,30]. Information on vaccination practices was obtained through a dichotomic question on vaccination against influenza within the past year (yes or no) and multi-choice questions assessing their opinion on influenza vaccine usefulness during pregnancy and breastfeeding, the fear of maternal and fetal/neonatal side effects, and overall vaccination acceptance.

### 2.4. Main Outcomes

COVID-19 vaccine willingness of pregnant and breastfeeding women if a vaccine had been available was evaluated through participants’ graded answers: “fully agree”, “rather agree”, “rather disagree”, or “fully disagree”. Participants who “fully agree” or “rather agree” were grouped as “willing to get vaccinated against SARS-CoV-2” and those who “rather disagree” or “fully disagree” were grouped as “not willing to get vaccinated against SARS-CoV-2” in the analysis.

### 2.5. Statistical Analysis

Baseline and medical characteristics, SARS-CoV-2 exposure (SARS-CoV-2 testing, symptoms, and hospitalization), fears, impacts of the pandemic, mental health symptoms, and vaccination habits were presented using descriptive statistics for both pregnant and breastfeeding women. The prevalence of participants willing to get vaccinated against COVID-19 was calculated.

The associations between variables of interest and the willingness to get vaccinated against SARS-CoV-2 was measured by univariate and multivariate logistic regression and were presented as crude odds ratios (OR) and adjusted odds ratios (aORs) with 95% confidence intervals (95% CI). Variables with *p* > 0.10 in the univariate analysis were not included in the multivariate model. The variables of interest were maternal age >40 years old, educational level (dichotomized as higher than high school or not), professional activity (dichotomized as active or not), primary language (French, German, Italian), maternal co-morbidities (grouped into a single “any maternal co-morbidity” variable), testing positive for SARS-CoV-2 infection (either by RT-PCR, serology, or CT-scan, grouped into a single “tested positive for SARS-CoV-2” variable), living with someone >65 years old, having a negative impact by the pandemic on the pregnancy/breastfeeding experience, life habits, and work, experiencing symptoms of severe depression (EDS ≥ 13), anxiety (GAD-7 ≥ 15), or high stress perceived (PSS ≥ 27) (grouped into a single variable), being vaccinated against influenza in the past year, previous history of declining vaccination, and fear of side effects related to vaccines (for the mother and the fetus/neonate). Variables specific to pregnant women (pregnancy practitioner, current trimester of gestation, and fear of an adverse fetal outcome in case of maternal SARS-CoV-2 infection) were studied in a supplementary multivariate model including only pregnant participants.

### 2.6. Missing Values

Maternal comorbidities were considered as absent if not reported, based on the assumption that severe comorbidities are normally documented. Based on the hypothesis of missing variables completely at random (MCAR), multiple imputations with chained equations (10 replications) were performed to increase the power of comparisons for missing values.

## 3. Results

A total of 2064 respondents participated in the survey (1161 using the French questionnaire, 868 using the German questionnaire, and 35 using the Italian questionnaire) including 1501 breastfeeding and 563 pregnant women. Among them, 513 (24.9%) did not answer the question relating to whether they were willing to get vaccinated against SARS-CoV-2 if a vaccine was available. Thus, 75.1% (*n* = 1551) contributed to the analyses addressing the primary aim of the study (1036 breastfeeding mothers and 515 pregnant women) (Figure 1).

### 3.1. Baseline Characteristics

Baseline characteristics are presented in Table 1. The median age of respondents was 33 years and the majority were married or cohabiting (79.8%; 1237/1551). A significant proportion of women were healthcare providers (20.4%; 317/1551) or homemakers (9.0%; 139/1551). A high proportion of participants (46.5%; 721/1551) had an education level above high school. Overall, 9.7% (151/1551) reported having co-morbidities.

Among the pregnant participants, the median gestational age was 28 weeks’ gestation at the time of survey completion. Half of them were multigravida (275/515), among which 74.4% (204/274) and 18.2% (50/274) had one or more previous children respectively. More than 90% (468/515) were under the care of an obstetrician. Among the breastfeeding participants, 2.8% (29/1036) had their neonates hospitalized in an intensive care unit.

### 3.2. SARS-CoV-2 Exposure, Fears, and Beliefs

Data on SARS-CoV-2 exposure, fears, and beliefs are presented in Table 2. Almost 55% (850/1551) of participants reported having experienced symptoms potentially related to SARS-CoV-2 within the 3 months preceding the survey. Only 10.9% (170/1551) of the women had been tested for SARS-CoV-2 infection, among which 10.5% had a positive result (18/170) through a PCR-based nasopharyngeal swab, serology, or CT-scan. Less than 1.0% (9/1551) reported having been hospitalized due to COVID-19. Only 1.2% (18/1551) of participants reported living with someone older than 65 years old. Participants reported that the COVID-19 pandemic had a negative impact on their pregnancy or breastfeeding experience in 35.3% (97/275) and 8.0% (41/512) of cases, respectively. According to their responses, 11.0% (170/1551) of them experienced symptoms of severe depression (EDS ≥ 13), anxiety (GAD-7 ≥ 15), or high stress (PSS ≥ 27) over the last four weeks. More than half of pregnant women (53.4%; 275/515) declared that they feared an adverse fetal outcome in case of maternal infection.

### 3.3. Vaccination Practices and Beliefs

Only 19.3% (85/440) of pregnant and 28.0% (249/891) of breastfeeding women were vaccinated against influenza in the past year. Among the participants, 10.5% (163/1551) and 0.1% (2/1551) mentioned fear of potential consequences for their fetus/infant or themselves respectively resulting from vaccination during pregnancy or breastfeeding. More than 20% (324/1551) of them indicated usually declining influenza vaccination, and 7.5% (117/1551) think the influenza vaccine is not needed during pregnancy or breastfeeding (Figure 2).

### 3.4. Willingness to Get the SARS-CoV-2 Vaccine

Only 29.7% (153/515) of pregnant and 38.6% (400/1036) of breastfeeding women were willing to get vaccinated against SARS-CoV-2 if a vaccine had been available during the first wave. More specifically, 8.1% (127/1551) fully agreed, 27.5% (426/1551) somewhat agreed, 40.4% (626/1551) somewhat disagreed, and 24% (372/1551) fully disagreed to get vaccinated (Figure 2 and Appendix A).

### 3.5. Factors Associated with SARS-CoV-2 Vaccine Willingness

Potential predictors of SARS-CoV-2 vaccine acceptance are shown in Table 3. Sociodemographic factors such as a maternal age above 40 years old (aOR 1.8 [1.1–3.2]), an educational level higher than high school (aOR 1.5 [1.2–2.0]), and Italian as a primary language (aOR 3.3 [1.4–8.0]) were associated with a higher rate of vaccine acceptance. On the other hand, German-speaking participants were less likely to get vaccinated (aOR 0.7 [0.5–0.9]).

Having had the influenza vaccination in the past year was a positive predictor for SARS-CoV-2 vaccine acceptance (aOR 2.1 [1.5–2.8]). Women who usually declined influenza vaccination were less likely to be willing to get the SARS-CoV-2 vaccine (aOR 0.2 [0.1–0.3]).

When assessing the impact of the SARS-CoV-2 pandemic, none of the variables showed statistically significant influence on the willingness to get vaccinated. However, a trend toward COVID-19 vaccine willingness can be observed among women having a positive diagnosis of SARS-CoV-2 (aOR 3.3 [0.8–13.7] and living with someone older than 65 years old (aOR 2.0 [0.7–6.1]).

Among the pregnant participants, those who had an obstetrician following their pregnancy (aOR 3.6 [1.2–11.2]) and who were in their third trimester of pregnancy (aOR 1.8 [1.1–2.7]) were more likely to be willing to receive the SARS-CoV-2 vaccine. On the other hand, being in their second trimester of pregnancy was associated with a higher SARS-CoV-2 vaccination refusal (aOR 0.6 [0.4–0.9]).

## 4. Discussion

Our results demonstrate that in Switzerland, only one-third (35.7%; 553/1551) of pregnant and breastfeeding women that participated in the survey were willing to get a SARS-CoV-2 vaccine during the first wave of the pandemic if one had been available. The positive predictors for SARS-CoV-2 vaccine acceptance among all participants were an age older than 40 years, a higher educational level, speaking Italian as their primary language, and having been vaccinated against influenza in the previous year. On the other hand, speaking German and usually declining influenza vaccination were negative predictors. Regarding pregnant participants, having an obstetrician following their pregnancy and being in their third trimester of pregnancy were two positive factors associated with the willingness to be vaccinated against SARS-CoV-2, whereas being in their second trimester of pregnancy was a negative predictor. No association was found between maternal co-morbidities and the participants’ willingness to get vaccinated.

### 4.1. Interpretation

Our study shows that despite Switzerland being among SARS-CoV-2 high incidence countries during the first wave with a particularly negative impact on pregnancy and breastfeeding experience [24], it has a low rate of SARS-CoV-2 vaccination acceptance. The results from our survey of Swiss women were among the lowest when compared to a recent survey conducted in 16 countries that showed a SARS-CoV-2 vaccine acceptance rate among pregnant women of 52.0%, with responses varying substantially between countries (28.8–84.4%) [31]. An American cross-sectional survey showed a rate of 41% of SARS-CoV-2 vaccine acceptance among pregnant women [32]. The low percentage of Swiss pregnant women willing to get the SARS-CoV-2 vaccine that we observed is consistent with the rather low influenza and pertussis immunization rates in Switzerland previously mentioned [16]. We also identified variability in SARS-CoV-2 vaccine acceptance between different regions of Switzerland. This has already been observed with Swiss-German women being more reluctant to get their children vaccinated [33,34]. In contrast, the part of Switzerland most affected by SARS-CoV-2, the Italian part, seems to have a higher rate of vaccine acceptance than the other parts of Switzerland, although fewer Italian speaking women were included, and these results should be interpreted with caution.

In this Swiss sub-analysis, the proportion of breastfeeding women willing to be vaccinated was higher than that of pregnant women (38.6% vs. 29.7%). This difference was also found in all countries included in the European study of which these data are a part, with a difference of up to +25% for the UK [25]. These results support that “vaccine hesitancy” may be even more common during pregnancy, which may be related to an overall greater reluctance to use medicines during pregnancy.

When assessing factors influencing SARS-CoV-2 vaccine willingness among pregnant women, our results are consistent with another study identifying older age and higher educational level as positive predictors [31]. The same observations have been made for acceptance of the pertussis and influenza vaccines [23,31]. The positive correlation that we observed between SARS-CoV-2 vaccine willingness and having received the influenza vaccine during the previous season has also been found in a recent study evaluating pregnant women [32]. In addition, we identified that being in the second trimester of pregnancy might be a negative predictor for SARS-CoV-2 vaccine acceptance, suggesting a potential fear for induced fetal malformations. This is consistent with several studies identifying fear for any potential harmful side effects of the vaccine on their fetus or infant as well as concerns regarding safety and effectiveness as major reasons for vaccine reluctance [22,31,32]. Concerns about teratogenicity would be more likely in the first trimester, as second trimester exposures do not cause embryopathy. Here, patients who responded to the survey as being in the second trimester correspond to those who were in the first trimester at the time of the first wave, so our results may suggest that their fear of teratogenicity may be higher in early pregnancy and not necessarily in the second trimester. In contrast, we did not find an association between having experienced symptoms of severe depression, anxiety, or high stress in the weeks prior to the survey and the willingness to get the SARS-CoV-2 vaccine. This is not in line with a previous study where pregnant women with a history of major depressive disorder and moderate anxiety were significantly more likely to get influenza and pertussis vaccines [23]. However, at the time of the survey, no SARS-CoV-2 vaccine was yet available, and thus, no information was available regarding its safety or effectiveness in general and in the pregnant population, which may explain the reluctance of anxious or depressed women who may need safety information before accepting the vaccine. This might also have influenced participants who did not answer the question about SARS-CoV-2 vaccination acceptance, as the ambivalence toward this vaccine is still very strong. It is also interesting to note that while most participants showed acceptance of influenza vaccines, a much smaller percentage actually received it. This might question the access of Swiss breastfeeding and pregnant women to vaccines and how healthcare workers might play a role in it.

Our observations suggest that more than a specific reluctance toward the SARS-CoV-2 vaccine, it is one’s personal opinion on vaccination during pregnancy in general that might prevent Swiss pregnant and breastfeeding women from getting vaccinated. Hence, it is our hypothesis that SARS-CoV-2 and influenza or pertussis vaccines are avoided for similar reasons: mainly the lack of recommendation by healthcare professionals and the lack of compliance by pregnant women. Until the end of May 2021, Switzerland has made access to vaccines challenging, even for women that might want to be vaccinated, which could represent a barrier for vaccine acceptance. Furthermore, the ongoing debates over SARS-CoV-2 vaccines may have a negative influence on the willingness of pregnant women to become vaccinated. This emphasizes the need to improve access to vaccination for pregnant women as well as knowledge and acceptance of immunization during pregnancy among healthcare workers and pregnant/breastfeeding women.

### 4.2. Strengths and Limitations

In terms of temporality, our study explored the experience of Swiss pregnant women during the first wave of the SARS-CoV-2 pandemic. Our study included a large number of participants from different parts of Switzerland, was conducted in three official languages, and is the first to address the question of SARS-CoV-2 vaccine willingness in the country. Selection bias might have occurred as the proportion of participants who are professionally active and highly educated was higher than the general population of Swiss pregnant women [35,36]. This could have led to an increased vaccination acceptance rate, as highly educated women tend to have a higher acceptance of vaccination, which would mean that the vaccine willingness in the overall perinatal population might be even lower than that reported here. The survey was conducted online and, although most Swiss women have good access to the internet, those that rely more on online resources may have come across the online survey more often when looking for information about their pregnancy or breastfeeding. Women hospitalized or severely ill might not have had the opportunity to participate. This could have biased the association between SARS-CoV-2 exposure and the participants’ willingness to get vaccinated toward the null. In addition, as only 5% of women declared speaking another language in our survey, we might have an under representation of the immigrant population.

Another limitation might be the overrepresentation of French-speaking participants, which could be explained by the CHUV (Centre Hospitalier Universitaire Vaudois, university hospital of the largest French-speaking canton) leading the present study. Since some studies have shown an increased vaccination acceptance among the French-speaking part of Switzerland, this could have overestimated the rate of SARS-CoV-2 vaccine willingness in our study. Overestimation of SARS-CoV-2 vaccination acceptance could have also happened since a high percentage of participants were healthcare workers, more likely to be exposed to SARS-CoV-2 positive patients, and thus, more prone to being immunized.

Factors reported to be associated with SARS-CoV-2 vaccine willingness, considered in other studies, were not measured [31,32]. Those include socioeconomic status; perceived risk of SARS-CoV-2 (likelihood of infection, self or infant); opinion on the importance to public health to get a vaccine and for the majority of people to get vaccinated; compliance with preventive measures; monitoring of SARS-CoV-2 news and updates; trust and satisfaction with health authorities; as well as trust in science. Further surveys including those variables would be needed to better specify the factors influencing SARS-CoV-2 vaccination acceptance among Swiss pregnant women.

Finally, this survey was conducted at a time when no SARS-CoV-2 vaccine had yet been accepted by Swissmedic nor recommended for pregnant women. This could represent an important bias, since participants were asked if they would accept a potential vaccine without information about its efficiency and safety. Since this survey, the first randomized controlled trial of SARS-CoV-2 vaccination in pregnancy has been initiated [37]. Additionally, following the example of several other countries, the Swiss Society for Gynecology and Obstetrics (SSGO) along with the Federal Public Health Office (OFSP) has recommended, up until the end of May 2021, SARS-CoV-2 vaccination during the second and third trimester for pregnant women at high risk of developing complications or at high risk of exposure [38]. Recent studies also showed robust immune responses and efficient passage of antibodies to newborns after SARS-CoV-2 vaccination of pregnant women [39,40], unlike transplacental immunization through infected mothers, which seems to be less effective [41]. As new guidelines and more data on vaccinated pregnant women become available every day [42], willingness to become vaccinated might evolve, and new studies are urgently needed.

## 5. Conclusions

Our study suggests disappointing SARS-CoV-2 vaccine willingness among Swiss pregnant and breastfeeding women, emphasizing the need to identify and reduce barriers toward immunization. Inclusion of pregnant women in clinical trials, improving access to vaccines, and providing tailored information for pregnant and breastfeeding women, especially for those of younger age with a lower educational level, are crucially needed to protect them from SARS-CoV-2 and other viral threats ahead.

## Figures and Tables

**Figure 1 viruses-13-01199-f001:**
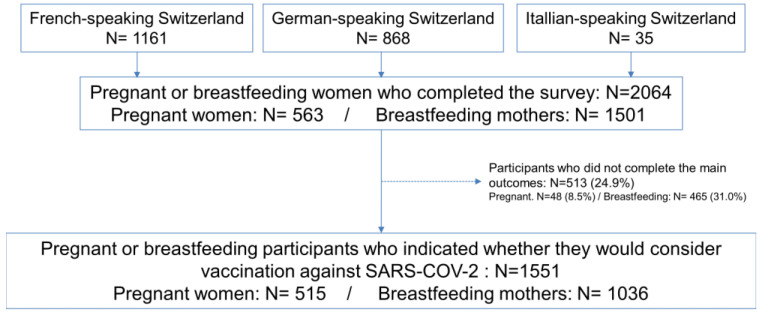
Flow chart.

**Figure 2 viruses-13-01199-f002:**
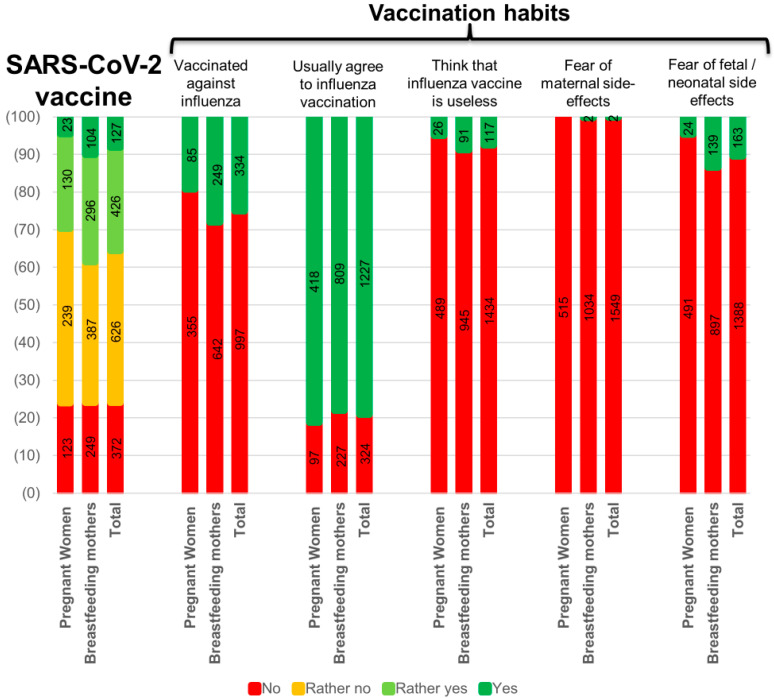
SARS-CoV-2 vaccine willingness among Swiss pregnant and breastfeeding women, and vaccination habits.

**Table 1 viruses-13-01199-t001:** Baseline characteristics and medical history of participants. Abbreviations: ENT, ear nose throat; IQR, interquartile range; NICU, neonatal intensive care unit, HCP, healthcare provider.

		Pregnant Women		Breastfeeding Mothers		Total	
		*n* = 515	(%)	*n* = 1036	(%)	*n* = 1551	(%)
Baseline characteristics							
Maternal age (years)—median (IQR)		33	(31–35)	33	(31–35)	33	(31–35)
	>40 years	19	(3.7)	63	(6.1)	82	(5.3)
Marital status							
	Married/cohabiting	422	(81.9)	815	(78.7)	1237	(79.8)
	Single/divorced/others	4	(0.8)	9	(0.9)	13	(0.8)
	Unknown	89	(17.3)	212	(20.5)	301	(19.4)
Working status							
	Health care provider	122	(23.7)	195	(18.8)	317	(20.4)
	Employed other than HCP	257	(49.9)	465	(44.9)	722	(46.6)
	Student	3	(0.6)	7	(0.7)	10	(0.6)
	Housewife	21	(4.1)	118	(11.4)	139	(9.0)
	Job seeker	12	(2.3)	23	(2.2)	35	(2.3)
	Unknown	100	(19.4)	228	(22.0)	328	(21.1)
Educational level							
	Less than high school	9	(1.8)	20	(1.9)	29	(1.9)
	High school	75	(14.6)	212	(20.5)	287	(18.5)
	More than high school	257	(49.9)	464	(44.8)	721	(46.5)
	Unknown	174	(33.8)	340	(32.8)	514	(33.0)
Primary language							
	French	217	(42.1)	418	(40.4)	635	(40.9)
	German	183	(35.5)	322	(31.1)	505	(32.6)
	Italian	8	(1.6)	23	(2.2)	31	(2.0)
	Other	18	(3.5)	61	(5.8)	79	(5.1)
	Unknown	89	(17.3)	212	(20.5)	301	(19.4)
Maternal co-morbidities							
Any comorbidity		51	(9.9)	100	(9.7)	151	(9.7)
	Pulmonary	14	(2.7)	28	(2.7)	42	(2.7)
	Cardio-vascular	6	(1.2)	11	(1.1)	17	(1.1)
	Pregestational diabetes	5	(1.0)	9	(0.9)	14	(0.9)
	Thyroid dysfunction	12	(2.3)	27	(2.6)	39	(2.5)
	Oncologic	1	(0.2)	2	(0.2)	3	(0.2)
	Hematologic	2	(0.4)	0	(0.0)	2	(0.1)
	Auto-immune	2	(0.4)	4	(0.4)	6	(0.4)
	Neurologic	3	(0.6)	4	(0.4)	7	(0.5)
	Psychic	3	(0.6)	6	(0.6)	9	(0.6)
	Digestive	3	(0.6)	7	(0.7)	10	(0.7)
	Uro-genital tract	6	(1.2)	15	(1.4)	21	(1.4)
	Cutaneous	2	(0.4)	4	(0.4)	6	(0.4)
	ENT	0	(0.0)	1	(0.1)	1	(0.1)
Smoking		69	(13.4)	149	(14.4)	218	(14.1)
Actual pregnancy or breastfeeding							
Practitioner:	Obstetrician	468	90.9	/			
	Midwife	13	8.3				
	Family physician	4	0.8				
Gestation	1	240	46.6	/			
	>1	275	53.4				
Parity	0	20/274	7.3	/			
	1	204/274	74.4				
	>1	50/274	18.2				
Planned pregnancy		483	93.8	/			
Gestational age—median (IQR)		28	(18-34)	/			
	1st Trimester	79	(15.0)				
	2nd Trimester	194	(40.7)				
	3rd Trimester	241	(44.3)				
Neonate hospitalized in NICU		/		29	(2.8)		

**Table 2 viruses-13-01199-t002:** SARS-CoV-2 exposure, fears, and beliefs. Abbreviations: PCR, polymerase chain reaction.

			Pregnant Women		Breastfeeding Mothers		Total	
			*n* = 515	(%)	*n* = 1036	(%)	*n* = 1551	(%)
SARS-COV-2 exposure								
Symptoms during the 3 last months			296	(57.5)	554	(53.5)	850	(54.8)
Hospitalized for COVID-19			2	(0.4)	7	(0.7)	9	(0.6)
Tested for SARS-CoV-2 infection			48	(9.3)	122	(11.8)	170	(10.9)
	PCR on nasopharyngeal swab		39	(7.6)	112	(108.0)	151	(9.7)
		positive	5/39	(12.8)	6/112	(5.3)	11/151	(7.3)
		negative	33/39	(84.6)	103/112	(92.0)	136/151	(90.1)
		unknown	1/39	(2.6)	3/112	(2.7)	4/151	(2.7)
	Serology		7	(1.4)	21	(2.0)	28	(1.8)
		positive	3/7	(42.9)	2/21	(9.5)	5/28	(17.9)
		negative	3/7	(42.9)	16/21	(76.2)	19/28	(67.9)
		unknown	1/7	(14.2)	3/21	(14.3)	4/28	(14.3)
	Scanner		2	(0.4)	2	(0.2)	4	(2.6)
		positive	2/2	(100.0)	0/2	(0.0)	2/4	(50.0)
		negative	0/2	(0.0)	2/2	(100.0)	2/4	(50.0)
Living with someone with symptoms			82	(15.9)	220	(21.2)	302	(19.5)
Living with someone tested positive			4	(0.8)	10	(1.0)	14	(0.9)
Living with someone > 65 years old			6	(1.2)	12	(1.2)	18	(1.2)
Negative impact of the COVID-19 pandemic on:								
Pregnancy or breastfeeding experience			97	(18.8)	41	(4.0)	138	(8.9)
	unknown		240	(46.6)	524	(50.6)	764	(49.3)
Life habits			350	(68.2)	700	(67.6)	1050	(67.7)
	unknown		8	(1.6)	25	(2.4)	33	(2.1)
Work			295	(57.3)	394	(38.0)	689	(44.4)
	unknown		100	(19.4)	320	(30.9)	420	(27.1)
Fear of an adverse fetal outcome			275	(53.4)	/			
Symptoms of severe depression, anxiety or high stress perceived during the 1st wave			53	(10.3)	117	(11.3)	170	(11.0)

**Table 3 viruses-13-01199-t003:** Factors associated with SARS-CoV-2 vaccine willingness among Swiss pregnant and breastfeeding women. Abbreviations: aOR, adjusted odds ratio; CI, confidence interval; CT, computed tomography; OR, odds ratio; RT-PCR, reverse-transcriptase polymerase chain reaction; T, trimester of gestation.

		Participants Willing to Get Vaccinated against COVID-19		Participants Not Willing to Get Vaccinated against COVID-19		OR	(95% CI)	*p*	aOR	(95% CI)	*p*
		N	(%)	N	(%)						
		553	(35.7)	998	(64.3)						
Baseline characteristics											
Maternal age >40 years		42	(7.6)	40	(4.0)	2.0	(1.3–3.0)	0.003	1.8	(1.1–3.2)	0.028
Educational level > highschool		300 *	(75.9)	421 *	(65.6)	1.7	(1.3–2.2)	<0.001	1.5	(1.1–2.0)	0.017
Professionally active		387 *	(87.4)	652 *	(83.6)	1.4	(1.0–1.9)	0.007	1.0	(0.7–1.5)	0.919
Primary language											
	French	238	(52.8)	397	(49.7)	1.1	(0.9–1.4)	0.295			
	German	159 *	(35.3)	346 *	(43.3)	0.7	(0.6–0.9)	0.005	0.7	(0.5–0.9)	0.015
	Italian	19 *	(4.2)	12 *	(1.5)	2.9	(1.4–6.0)	0.005	3.3	(1.4–8.0)	0.007
Any maternal co-morbidity		58	(10.5)	93	(9.3)	1.1	(0.8–1.6)	0.457			
Impact of the SARS-COV-2 pandemic											
Tested positive for SARS-COV-2 (RT-PCR, serology and/or CT)		9	(1.6)	3	(0.3)	5.5	(1.5–20.4)	0.011	3.3	(0.8–13.7)	0.095
Living with someone >65 years old		10	(1.8)	8	(0.8)	2.3	(0.8–6.7)	0.076	2.0	(0.7–6.1)	0.094
Negative impact of the pandemic on											
	Pregnancy	52	(19.9)	86	(16.3)	1.1	(1.0–1.2)	0.215			
	Life habits	398 *	(72.8)	652 *	(67.2)	1.3	(1.0–1.7)	0.023	1.0	(0.8–1.4)	0.822
	Work	244	(60.1)	445	(61.4)	1.0	(0.7–1.2)	0.672			
Symptoms of severe depression, anxiety or high stress		68	(12.3)	102	(10.2)	1.2	(0.9–1.7)	0.211			
Vaccination habits and beliefs											
Vaccinated against Influenza last year		197 *	(41.1)	137 *	(16.1)	3.6	(2.8–4.7)	<0.001	2.1	(1.5–2.8)	<0.001
Usually decline vaccination		30	(5.4)	294	(29.5)	0.1	(0.1–0.2)	<0.001	0.2	(0.1–0.3)	<0.001
Fear of side effects related to vaccines		51	(9.2)	114	(11.4)	0.8	(0.6–1.1)	0.179			
Supplementary model including pregnancy-related variables *(tested only in pregnant women, N = 515)*		N	(%)	N	(%)	OR	(95%CI)	*p*	aOR	(95%CI)	*p*
		153	(29.7)	362	(60.3)						
Follow-up by an obstetrician		144	(94.1)	324	(89.5)	1.9	(0.9–4.0)	0.101	3.6	(1.2–11.2)	0.027
Gestational age											
	T1	25	(16.3)	54	(15.0)	1.1	(0.7–1.9)	0.691			
	T2	47	(30.7)	147	(40.7)	0.6	(0.4–1.0)	0.033	0.6	(0.4–0.9)	0.015
	T3	81	(52.9)	160	(44.3)	1.4	(1.0–2.0)	0.074	1.8	(1.1–2.7)	0.018
Fear of an adverse fetal outcome in case of infection		75	(49.0)	200	(55.3)	0.9	(0.8–1.0)	0.196			

* Multiple imputations on missing values.

## Data Availability

The data presented in this study are available on request from the corresponding author.

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
