# Peer review of "SARS-CoV-2 Vaccine Willingness among Pregnant and Breastfeeding Women during the First Pandemic Wave: A Cross-Sectional Study in Switzerland"

_viruses, 2021, doi:10.3390/v13071199_

Round 1
Reviewer 1 Report
In general, the topic is of great importance. Vaccine hesitancy has been increasing recently, even in the pre-covid era, causing outbreaks of preventable diseases. Any new information related to this topic is of much importance.
Introduction: well written, with appropriate references.
Methods: an approval of the appropriate review board should be included, or stated if it was not needed and why? How was eligibility checked? A CT scan is not an appropriate study to test for COVID-19.
Results: It seems that a high rate of participants (24.9%) did not provide an answer for the main study objective (ie acceptance of the COVID-19 vaccine). Do the authors think this is relevant, and what the reason for this is? Similarly, a high proportion of the participants were health care workers, which should be discussed. The presentation of the results, including the graphs, is excellent. It is interesting to see that while most participants showed acceptance of influenza vaccines, a much smaller percentage actually received it. This would deserve a discussion as well.
Discussion: the authors seem to discuss pregnant and breast feeding participants as one group, although it should be two different groups. Please do not use the term "assume" in the discussion. Using "we theorize" or "it is our hypothesis" is more appropriate.
The discussion on limitations is excellent. My only recommendation here is to include a discussion of the high percentage of health care workers among the participants, and to discuss the potential reasons why such high percentage did not respond to the main objective of the study.
Author Response
In general, the topic is of great importance. Vaccine hesitancy has been increasing recently, even in the pre-covid era, causing outbreaks of preventable diseases. Any new information related to this topic is of much importance.
Introduction: well written, with appropriate references.
We thank the reviewer for their very positive comment and for their help to improve our manuscript.
Methods: an approval of the appropriate review board should be included, or stated if it was not needed and why? How was eligibility checked? A CT scan is not an appropriate study to test for COVID-19.
We have modified the methods to read:
Line 83 “This Swiss cross-sectional online study is part of a European multi-center study conducted in several countries (Belgium, Ireland, Norway, The Netherlands, United Kingdom) and approved by the Ethics Committee Research of UZ/KU Leuven (id: S63966)”
We did not need an additional local ethical agreement for the sub-analysis of the Swiss data.
Eligibility was controlled at the beginning of the survey by questions on maternal age, whether or not they were currently pregnant or breastfeeding in the three previous months.
The methods include the following statement: “To be eligible, Swiss women needed to be at least 18 years old and be pregnant at the time of the survey or have breastfed within the past three months”.
During the first wave of the pandemic, PCR tests may have been difficult to access, and based on specific CT imaging we also preferred to ask participants if they had had these findings highly suggestive of COVID-19. A recent Cochrane systematic review found that the sensitivity of chest CT for the diagnosis of COVID-19 is 88% (https://www.cochrane.org/CD013639/INFECTN_how-accurate-chest-imaging-diagnosing-covid-19). Nevertheless, our 2 participants with a positive CT scan also had positive PCR results, not compromising the interpretation of our results. We therefore prefer to leave the information on the CT results for more transparency in testing.
Results: It seems that a high rate of participants (24.9%) did not provide an answer for the main study objective (ie acceptance of the COVID-19 vaccine). Do the authors think this is relevant, and what the reason for this is?
Since “I don’t know” was not an answer offered to the participants, it might have dissuaded some undecided ones not to answer at all. This could be relevant if put into perspective with the important overall population ambivalence towards vaccination especially for SARS-CoV-2.
See modifications under "Discussion"
Similarly, a high proportion of the participants were health care workers, which should be discussed.
See modifications under "Discussion"
The presentation of the results, including the graphs, is excellent.
We thank the reviewer for their rewarding comment.
It is interesting to see that while most participants showed acceptance of influenza vaccines, a much smaller percentage actually received it. This would deserve a discussion as well.
See modification under "Discussion"
Discussion: the authors seem to discuss pregnant and breast feeding participants as one group, although it should be two different groups.
To fulfill this comment, we have included a specific paragraph for breastfeeding women in the interpretation:
Line 272: “In this Swiss sub-analysis, the proportion of breastfeeding women willing to be vaccinated was higher than that of pregnant women (38.6% vs 29.7%). This difference was also found in all countries included in the European study of which these data are a part, with a difference of up to +25% for the UK. These results support that “vaccine hesitancy” may be even more common during pregnancy, which may be related to an overall greater reluctance to use medicines during pregnancy.”
Please do not use the term "assume" in the discussion. Using "we theorize" or "it is our hypothesis" is more appropriate.
We have modified to read:
Line 349: "Hence, it is our hypothesis that SARS-CoV-2 and influenza or pertussis vaccines are avoided for similar reasons: mainly the lack of recommendation by healthcare professionals and the lack of compliance by pregnant women."
The discussion on limitations is excellent. My only recommendation here is to include a discussion of the high percentage of health care workers among the participants, and to discuss the potential reasons why such high percentage did not respond to the main objective of the study.
We have modified as follows:
Line 388: "Overestimation of SARS-CoV-2 vaccination acceptance could have also happened since a high percentage of participants were healthcare workers, more likely to be exposed to SARS-CoV-2 positive patients, and thus, more prone to being immunized."
And:
Line 337: "This might also have influenced participants who didn’t answer the question about SARS-CoV-2 vaccination acceptance, as the ambivalence towards this vaccine is still very strong. It is also interesting to note that, while most participants showed acceptance of influenza vaccines, a much smaller percentage actually received it. This might question the access of Swiss breastfeeding and pregnant women to vaccines and how healthcare workers might play a role in it."
Reviewer 2 Report
In this paper the authors describe the COVID-19 willingness among pregnant and breastfeeding women if a vaccine was available from June 18th to July 12th in 2020. The low acceptance rate of the vaccine is surprising. The follow points are for authors’ considerations:
- Currently the COVID-19 vaccine is recommended for pregnant women who have a risk factor or are at high risk of exposure as health care providers. In this study 20.4% of participants were health care providers with high risk of SARS-CoV-2 exposure, and if the data is available, it could be interesting to compare the percentage of pregnant and breastfeeding women who actually are receiving the vaccine a year later of the questionnaire.
- The authors claim the low percentage of Swiss pregnant women willing to get the SARS-CoV-2 vaccine is according with immunizations rates for influenza and pertussis between pregnant women, possibly for similar reasons (lack of recommendation by healthcare professionals and lack of compliance by pregnant women). Do the authors have any thoughts about how to increase confidence in the vaccine with the pregnant population in Switzerland?
Author Response
Reviewer #2:
- Currently the COVID-19 vaccine is recommended for pregnant women who have a risk factor or are at high risk of exposure as health care providers. In this study 20.4% of participants were health care providers with high risk of SARS-CoV-2 exposure, and if the data is available, it could be interesting to compare the percentage of pregnant and breastfeeding women who actually are receiving the vaccine a year later of the questionnaire.
It is a great idea to investigate the percentage of Swiss pregnant and breastfeeding women who actually got vaccinated. This is why we launch a second multi-centric survey, covering the second and third pandemic waves from June 14, 2021 (https://kuleuven.eu.qualtrics.com/jfe/form/SV_06Y0QTpnlBLvHGm?fbclid=IwAR2yUjIEPGQK8jVwWjabckq3jhFoasgOkuW-3GI4ld7vjpZVaFV9VMD8IoE). This second survey will allow us to study the evolution in vaccine acceptance, and to compare the rate of pregnant and breastfeeding women having been vaccinated with those willing to be vaccinated.
- The authors claim the low percentage of Swiss pregnant women willing to get the SARS-CoV-2 vaccine is according with immunizations rates for influenza and pertussis between pregnant women, possibly for similar reasons (lack of recommendation by healthcare professionals and lack of compliance by pregnant women). Do the authors have any thoughts about how to increase confidence in the vaccine with the pregnant population in Switzerland?
In our conclusion, we suggest those improvement to increase confidence in the vaccine:
Line 422: "Inclusion of pregnant women in clinical trials, improving access to vaccines, and providing tailored information for pregnant and breastfeeding women, especially for those of younger age with a lower educational level, are crucially needed to protect them from SARS-CoV-2 and other viral threats ahead."